# Shape-Texture Debiased Neural Network Training

**Yingwei Li**[1]**, Qihang Yu**[1]**, Mingxing Tan**[2]**, Jieru Mei**[1]**, Peng Tang**[1]**, Wei Shen**[3]
**Alan Yuille**[1] **& Cihang Xie**[4]

[1]Johns Hopkins University  [2]Google Brain  [3]Shanghai Jiaotong University
[4]University of California, Santa Cruz

## Abstract

Shape and texture are two prominent and complementary cues for recognizing objects. Nonetheless, Convolutional Neural Networks are often biased towards either texture or shape, depending on the training dataset. Our ablation shows that such bias degenerates model performance. Motivated by this observation, we develop a simple algorithm for shape-texture debiased learning. To prevent models from exclusively attending on a single cue in representation learning, we augment training data with images with conflicting shape and texture information (*e.g.*, an image of chimpanzee shape but with lemon texture) and, *most importantly, provide the corresponding supervisions from shape and texture simultaneously*.

Experiments show that our method successfully improves model performance on several image recognition benchmarks and adversarial robustness. For example, by training on ImageNet, it helps ResNet-152 achieve substantial improvements on ImageNet (+1.2%), ImageNet-A (+5.2%), ImageNet-C (+8.3%) and Stylized-ImageNet (+11.1%), and on defending against FGSM adversarial attacker on ImageNet (+14.4%). Our method also claims to be compatible to other advanced data augmentation strategies, *e.g.*, Mixup and CutMix. The code is available here: `https://github.com/LiYingwei/ShapeTextureDebiasedTraining`.

## 1 Introduction

It is known that both shape and texture serve as essential cues for object recognition. A decade ago, computer vision researchers had explicitly designed a variety of hand-crafted features, either based on shape (*e.g.*, shape context (Belongie et al., 2002) and inner distance shape context (Ling & Jacobs, 2007)) or texture (*e.g.*, textons (Malik et al., 2001)), for object recognition. Moreover, researchers found that properly combining shape and texture can further recognition performance (Shotton et al., 2009; Zheng et al., 2007), demonstrating the superiority of possessing both features.

Nowadays, as popularized by Convolutional Neural Networks (CNNs) (Krizhevsky et al., 2012), the features used for object recognition are automatically learned, rather than manually designed. This change not only eases human efforts on feature engineering, but also yields much better performance on a wide range of visual benchmarks (Simonyan & Zisserman, 2015; He et al., 2016; Girshick et al., 2014; Girshick, 2015; Ren et al., 2015; Long et al., 2015; Chen et al., 2015). But interestingly, as pointed by Geirhos et al. (2019), the features learned by CNNs tend to bias toward either shape or texture, depending on the training dataset.

We verify that such biased representation learning (towards either shape or texture) weakens CNNs' performance.[1] Nonetheless, surprisingly, we also find (1) the model with shape-biased representations and the model with texture-biased representations are highly complementary to each other, *e.g.*, they focus on completely different cues for predictions (an example is provided in Figure 1); and (2) being biased towards either cue may inevitably limit model performance, *e.g.*, models may not be able to tell the difference between a lemon and an orange without texture information. These observations altogether deliver a promising message—biased models (*e.g.*, ImageNet trained (texture-biased) CNNs (Geirhos et al., 2019) or (shape-biased) CNNs (Shi et al., 2020)) are improvable.

---

[1]Biased models are acquired similar to Geirhos et al. (2019), see Section 2 for details.

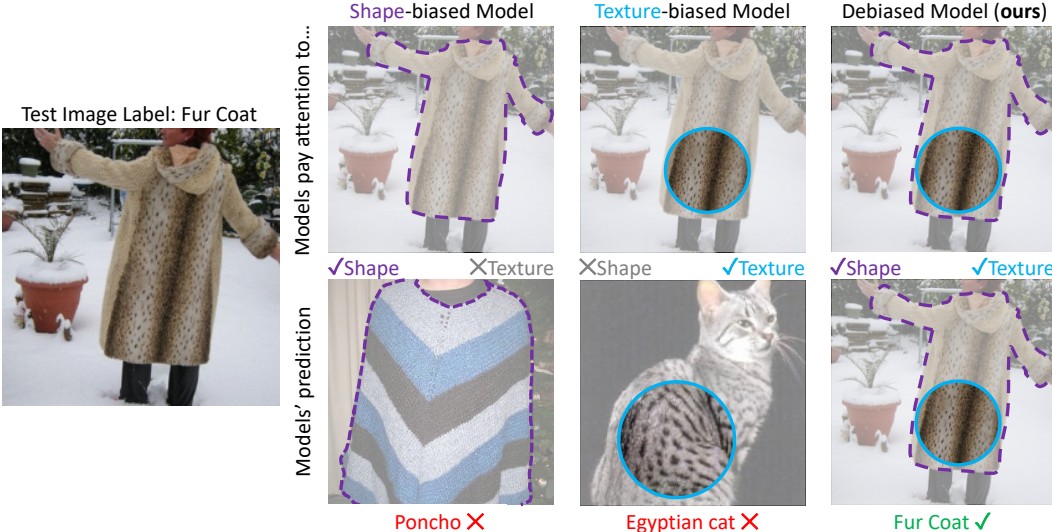

Figure 1: Both shape and texture are essential cues for object recognition, and biasing towards either one degenerates model performance. As shown above, when classifying this fur coat image, the shape-biased model is confounded by the cloth-like shape therefore predict it as a *poncho*, and the texture-biased model confuses it as an *Egyptian cat* because of the misleading texture. Nonetheless, our debiased model can successfully recognize it as a *fur coat* by leveraging both shape and texture.

To this end, we hereby develop a shape-texture debiased neural network training framework to guide CNNs for learning better representations. Our method is a data-driven approach, which let CNNs automatically figure out how to avoid being biased towards either shape or texture from their training samples. Specifically, we apply style transfer to generate cue conflict images, which breaks the correlation between shape and texture, for augmenting the original training data. The most important recipe of training a successful shape-texture debiased model is that we need to provide supervision from both shape and texture on these generated cue conflict images, otherwise models will remain being biased.

Experiments show that our proposed shape-texture debiased neural network training significantly improves recognition models. For example, on the challenging ImageNet dataset (Russakovsky et al., 2015), our method helps ResNet-152 gain an absolute improvement of 1.2%, achieving 79.8% top-1 accuracy. Additionally, compared to its vanilla counterpart, this debiased ResNet-152 shows better generalization on ImageNet-A (Hendrycks et al., 2019) (+5.2%), ImageNet-C (Hendrycks & Dietterich, 2019) (+8.3%) and Stylized ImageNet (Geirhos et al., 2019) (+11.1%), and stronger robustness on defending against FGSM adversarial attacker on ImageNet (+14.4%). Our shape-texture debiased neural network training is orthogonal to other advanced data augmentation strategies, *e.g.*, it further boosts CutMix-ResNeXt-101 (Yun et al., 2019) by 0.7% on ImageNet, achieving 81.2% top-1 accuracy.

## 2 SHAPE/TEXTURE BIASED NEURAL NETWORKS

The biased feature representation of CNNs mainly stems from the training dataset, *e.g.*, Geirhos et al. (2019) point out that models will be biased towards shape if trained on Stylized-ImageNet dataset. Following Geirhos et al. (2019), we hereby present a similar training pipeline to acquire shape-biased models or texture-biased models. By evaluating these two kinds of models, we observe the necessity of possessing both shape and texture representations for CNNs to better recognize objects.

### 2.1 MODEL ACQUISITION

**Data generation.** Similar to Geirhos et al. (2019), we apply images with conflicting shape and texture information as training samples to obtain shape-biased or texture-biased models. But different from Geirhos et al. (2019), an important change in our cue conflict image generation procedure is that we override the original texture information *with the informative texture patterns from another*

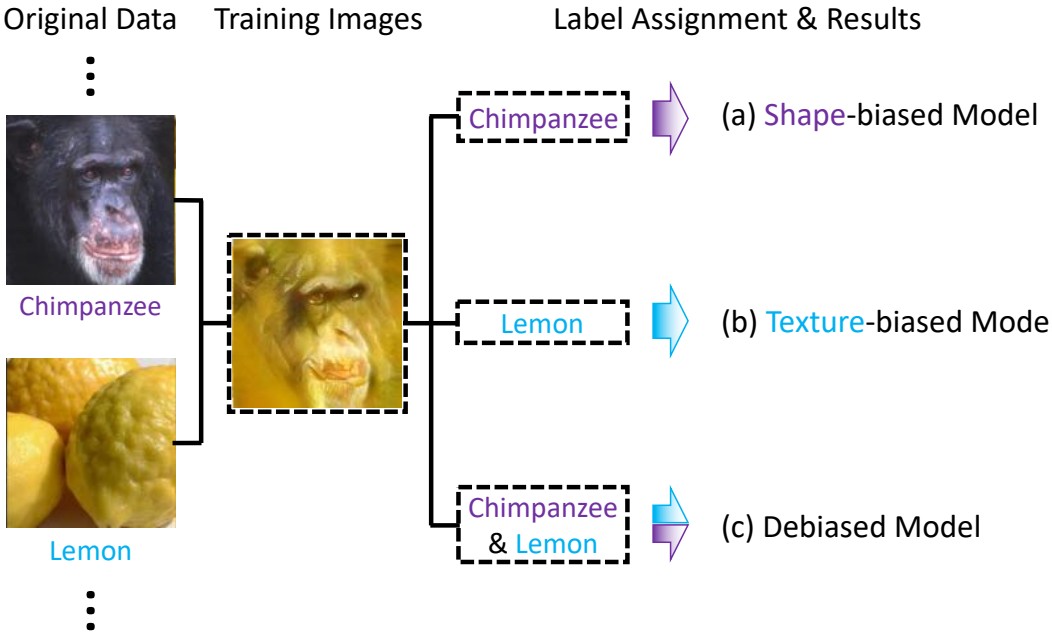

Figure 2: Illustration of the our training pipeline for acquiring (a) a shape-biased model, (b) a texture-biased model, and (c) a shape-texture debiased model. Specifically, these models share the same training samples, *i.e.* images with conflicting texture and shape information, generated by style transfer between two randomly selected images; but apply distinct labelling strategies: in (a) & (b), labels are determined by the images that provides shape (or texture) information in style transfer, for guiding models to learn more shape (or texture) representations; in (c), labels are jointly determined by the pair of images in style transfer, for avoiding bias in representation learning.

*randomly selected image, rather than with the uninformative style of randomly selected artistic paintings.* That being said, to create a new training sample, we need to first select a pair of images from the training set uniformly at random, and then apply style transfer to blend their shape and texture information. Such a generated example is shown in Figure 2, *i.e.*, the image of chimpanzee shape but with lemon texture.

**Label assignment.** The way of assigning labels to cue conflict images controls the bias of learned models. Without loss of generality, we show the case of learning a texture-biased model. To guide the model to attend more on texture, the labels assigned to the cue conflict images here will be exclusively based on the texture information, *e.g.*, *the image of chimpanzee shape but with lemon texture will be labelled as lemon*, shown in Figure 2(b). By this way, the texture information is highly related to the "ground-truth" while the shape information only serves as a nuisance factor during learning. Similarly, to learn a shape-biased model, the label assignment of cue conflict images will be based on shape only, *e.g.*, *the image of chimpanzee shape but with lemon texture now will be labelled as chimpanzee*, shown in Figure 2(a).

## 2.2 EVALUATION AND OBSERVATION

To reduce the computational overhead in this ablation, all models are trained and evaluated on ImageNet-200, which is a 200 classes subset of the original ImageNet, including 100,000 images (500 images per class) for training and 10,000 images (50 images per class) for validation. Akin to Geirhos et al. (2019), we observe that the models with biased feature representations tend to have inferior accuracy than their vanilla counterparts. For example, our shape-biased ResNet-18 only achieves 73.9% top-5 ImageNet-200 accuracy, which is much lower than the vanilla ResNet-18 with 88.2% top-5 ImageNet-200 accuracy.

Though biased representations weaken the overall classification accuracy, surprisingly, we find they are highly complementary to each other. We first visualize the attended image regions of biased models, via Class Activation Mapping (Zhou et al., 2016), in Figure 3. As we can see here, the shape-biased model and the texture-biased model concentrate on different cues for predictions. For

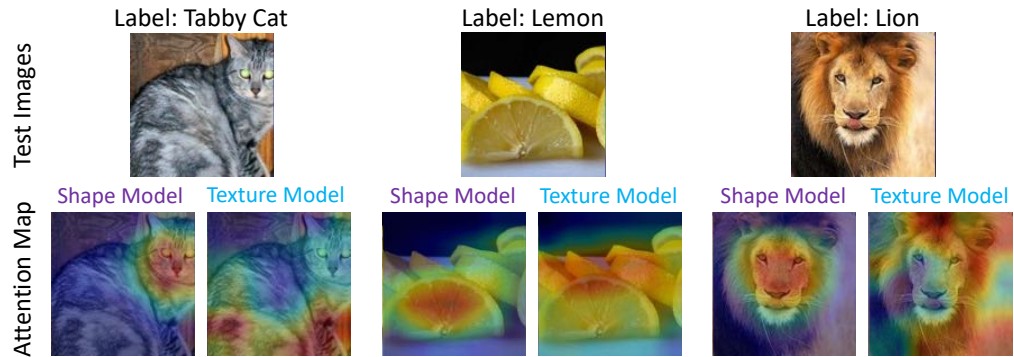

Figure 3: The shape-biased model and the texture-biased model attend on complementary cues for predictions. We use Class Activation Mapping to visualize which image regions are attended by models. Redder regions indicates more attentions are paid by models.

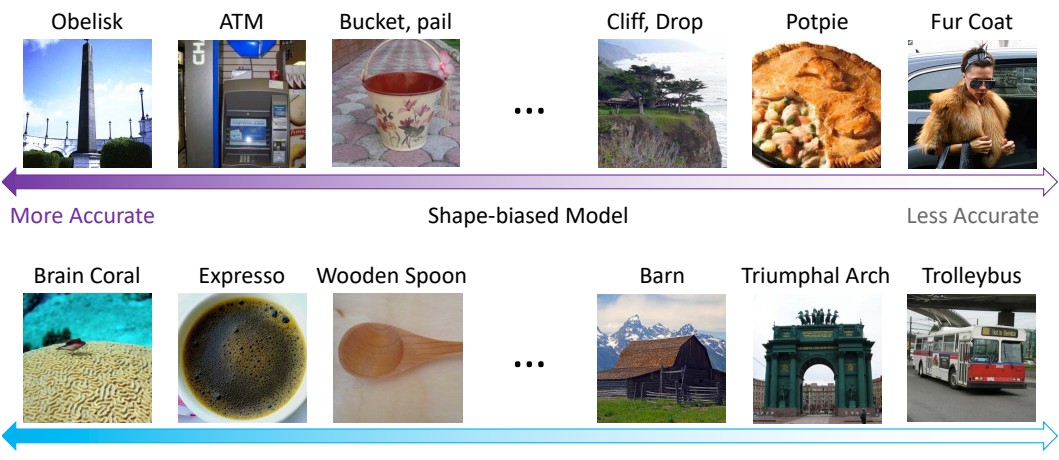

Figure 4: The shape-biased model and the texture-biased model are good/bad at classifying different object categories. We sort these object categories according to the model's corresponding top-1 accuracy, where the righter one indicates a lower accuracy achieved by the model.

instance, on the leftmost tabby cat image, the shape-biased model mainly focuses on the cat head, while the texture-biased model mainly focuses on the lower body and the front legs of the cat. Such attention mechanisms are correlated to their learned representations—the shape-biased model extracts the shape of the cat head as an important signal for predictions, while the texture-biased model relies on the texture information of cat fur for predictions.

As distinct cues are picked by shape-biased/texture-biased models, a more concrete observation is they are good/bad at classifying quite different object categories. As showed in Figure 4, the shape-biased model is good at recognizing objects with representative shape structure like *obelisk*, but is bad at recognizing objects whose shape is uninformative or almost indistinguishable from others like *fur coat*. Similarly, the texture-biased model can effectively recognize objects with unique texture patterns like *brain coral* but may fail to recognize objects with unpredictable texture like *trolleybus* (as its side body can be painted with different advertisements). Besides, biased models may inevitably perform poorly on certain categories as insufficient cues are applied. For examples, it is challenging to distinguish between a lemon and an orange if texture information cannot be utilized, or to distinguish between an lion and a tabby cat without shape information.

Given the analysis above, we can conclude that biased representations limit models' recognition ability. But meanwhile, our ablation delivers a promising message—the features learned by biased models are highly complementary to each other. This observation indicates the current training framework is improvable (as the resulted models are biased towards texture (Geirhos et al., 2019) or shape (Shi et al., 2020)), and offers a potential direction for building a stronger one—we should train models to properly acquire both shape and texture feature representations. We will introduce a simple method for doing so next.

## 3 Shape-Texture Debiased Neural Network Training

Recall that when obtaining a biased model, the strategy of label assignment is pivot—when the labels are exclusively determined by the images that provide shape (or texture) information in style transfer, we will obtain a shape-biased (or texture-biased) model. Therefore, to guide models for leveraging both shape and texture for predictions, we hereby propose a simple way, which is inspired by Mixup (Zhang et al., 2018), to softly construct labels during training. In other words, given the one-hot label of the shape-source image $y_s$ and the one-hot label of the texture-source image $y_t$, the new label that we assigned to the cue conflict image is

$$\widetilde{y} = \gamma * y_s + (1 - \gamma) * y_t, \tag{1}$$

where $\gamma \in [0, 1]$ is a manually selected hyperparameter to control the relative importance between shape and texture. By ranging the shape-texture coefficient $\gamma$ from 0 to 1, we obtain a path to evolve the model from being a texture-biased one (*i.e.*, $\gamma = 0$) to being a shape-biased one (*i.e.*, $\gamma = 1$). Although the two extreme ends lead to biased models with inferior performance, we empirically show that there exist a sweet point along this interpolation path, *i.e.*, the learned models can properly acquires both shape and texture feature representations and achieve superior performance on a wide range of image recognition benchmarks.

We name this simple method as shape-texture debiased neural network training, and illustrate the training pipeline in Figure 2(c). It is worth to mention that, although Figure 2 only shows the procedure of applying our method to the image classification task, this training framework is general and has the potential to be extended to other computer vision tasks, *e.g.*, a simple showcase on semantic segmentation is presented in Section 4.4.

## 4 Experiments

### 4.1 Experiments Setup

**Datasets.** We evaluate models on ImageNet classification and PASCAL VOC semantic segmentation. *ImageNet* dataset (Russakovsky et al., 2015) consists of 1.2 million images for training, and 50,000 for validation, from 1,000 classes. *PASCAL VOC* 2012 segmentation dataset (Everingham et al., 2012) with extra annotated images from (Hariharan et al., 2011) involves 20 foreground object classes and one background class, including 10,582 training images and 1,449 validation images.

Going beyond the standard benchmarks, we further evaluate models' generalization on ImageNet-A, ImageNet-C and Stylized-ImageNet, and robustness by defending against FGSM adversarial attacker on ImageNet. *ImageNet-C* (Hendrycks & Dietterich, 2019) is a benckmark dataset that measures models' corruption robustness. It is constructed by applying 75 common visual corruptions to the ImageNet validation set. *ImageNet-A* (Hendrycks et al., 2019) includes 7,500 natural adversarial examples that successfully attacks unseen classifiers. These examples are much harder than original ImageNet validation images due to scene complications encountered in the long tail of scene configurations and by exploiting classifier blind spots (Hendrycks et al., 2019). *Stylized-ImageNet* (Geirhos et al., 2019) is a stylized version of ImageNet that constructed by re-rendering the original images by AdaIN stylizer (Huang & Belongie, 2017). The generated images keep the original global shape information but removes the local texture information. *FGSM* (Goodfellow et al., 2015) is a widely used adversarial attacker to evaluate model robustness. We set the maximum perturbation change per pixel $\epsilon = 16/255$ for FGSM.

**Implementation details.** We choose ResNet (He et al., 2016) as the default architecture. For image classification tasks, our implementation is based on the publicly available framework in PyTorch[2]. To generate cue conflict images, we follow Geirhos et al. (2019) to use Adaptive Instance Normalization (Huang & Belongie, 2017) in style transfer, and set stylization coefficient $\alpha = 0.5$. Importantly, to increase the diversity of training samples, we generate these cue conflict images on-the-fly during training. We choose the shape-texture coefficient $\gamma = 0.8$ when assigning labels.

When training shape-biased, texture-biased and our shape-texture debiased models, we always apply the auxiliary batch normalization (BN) design (Xie et al., 2020; Xie & Yuille, 2020; Chen et al.,

---

[2]`https://github.com/bearpaw/pytorch-classification`

| | VANILLA | 2×EPOCHS | S-BIASED | T-BIASED | DEBIASED |
|---|---|---|---|---|---|
| ResNet-50 | 76.4 | 76.4 (+0.0) | 76.2 (-0.2) | 75.3 (-1.1) | 76.9 (+0.5) |
| ResNet-101 | 78.0 | 78.0 (+0.0) | 78.0 (-0.0) | 77.4 (-0.6) | 78.9 (+0.9) |
| ResNet-152 | 78.6 | 79.1 (+0.5) | 78.6 (-0.0) | 78.1 (-0.5) | 79.8 (+1.2) |

Table 1: The performance of the vanilla training, the shape-biased (S-biased) training, the texture-biased (T-biased) training, and our shape-texture debiased training on ImageNet. For all ResNet models, our debiased training shows the best performance among others.

| | IN-A
Acc. ↑ | IN-C
mCE ↓ | S-IN
Acc. ↑ | FGSM
Acc. ↑ |
|---|---|---|---|---|
| ResNet-50 | 2.0 | 75.0 | 7.4 | 17.1 |
| + Debiased | 3.5 (+1.5) | 67.5 (-7.5) | 17.4 (+10.0) | 27.4 (+10.3) |
| ResNet-101 | 5.6 | 69.8 | 9.9 | 23.1 |
| + Debiased | 9.1 (+3.5) | 62.2 (-7.6) | 22.0 (+12.1) | 34.4 (+11.3) |
| ResNet-152 | 7.4 | 67.2 | 11.3 | 25.2 |
| + Debiased | 12.6 (+5.2) | 58.9 (-8.3) | 22.4 (+11.1) | 39.6 (+14.4) |

Table 2: The model robustness on ImageNet-A (IN-A), ImageNet-C (IN-C), Stylized-ImageNet (S-IN), and on defending against FGSM adversarial attacker on ImageNet. Our shape-texture debiased neural network training significantly boosts the model robustness over the vanilla training baseline.

2021) to bridge the domain gap between the original data and the augmented data, *i.e.*, the main BN is exclusively running on original ImageNet images and the auxiliary BN is exclusively running on cue conflict images. We follow Xie et al. (2020) to always apply the main BN for performance evaluation. Besides, since our biased models and debiased models are all trained with both the original data and the augmented data (*i.e.*, 2× data are used in training), we also consider a stronger baseline (*i.e.*, 2× epochs training) which doubles the schedule of the vanilla training baseline, for the purpose of matching the total training cost.

## 4.2 RESULTS

**Model accuracy.** Table 1 shows the results on ImageNet. For all ResNet models, the proposed shape-texture debiased neural network training consistently outperforms the vanilla training baseline. For example, it helps ResNet-50 achieve 76.9% top-1 accuracy, beating its vanilla counterpart by 0.5%. Our method works better for larger models, *e.g.*, it further improves the vanilla ResNet-152 by 1.2%, achieving 79.8% top-1 accuracy.

We then compare our shape-texture debiased training to the 2× epochs training baseline. We find that simply doubling the schedule of the vanilla training baseline cannot effectively lead to improvements like ours. For examples, compared to the vanilla ResNet-101, this 2× epochs training fails to provide additional improvements, while ours furthers the top-1 accuracy by 1.0%. This result suggests that it is non-trivial to improve performance even if more computational budgets are given.

Lastly, we compare ours to the biased training methods. Though the only difference between our method and the biased training methods is the strategy of label assignment (as shown in Figure 2), it imperatively affects model performance. For example, compared to the vanilla baseline, both the shape-biased training and the texture-biased training fail to improve (sometimes even slightly hurt) the model accuracy, while our shape-texture debiased neural network training successfully leads to consistent and substantial accuracy improvements.

**Model robustness.** Next, we evaluate models' generalization on ImageNet-A, ImageNet-C and Stylized-ImageNet, and robustness on defending against FGSM on ImageNet. We note these tasks are much more challenging than the original ImageNet classification, *e.g.*, the ImageNet trained ResNet-50 only achieves 2.0% accuracy on ImageNet-A, 75.0% mCE on ImageNet-C, 7.4% accuracy on Stylized-ImageNet, and 17.1% accuracy on defending against FGSM adversarial attacker. As shown in Table 2, our shape-texture debiased neural network training beats the vanilla training baseline by a large margin on all tasks for all ResNet models. For example, it substantially boosts ResNet-152's performance on ImageNet-A (+5.2%, from 7.4% to 12.6%), ImageNet-C (−8.3%, from 67.2% to 58.9%, the lower the better) and Stylized-ImageNet (+11.1%, from 11.3% to 22.4%), and on defending against FGSM on ImageNet (+14.4%, from 25.2% to 39.6%). These results altogether suggest that our shape-texture debiased neural network training is an effective way to mitigate the issue of shortcut learning (Geirhos et al., 2020).

|  | IN Acc. ↑ | IN-A Acc. ↑ | IN-C mCE ↓ | S-IN Acc. ↑ | FGSM Acc. ↑ |
|---|---|---|---|---|---|
| ResNet-50 | 76.4 | 2.0 | 75.0 | 7.4 | 17.1 |
| CutMix + MoEx (Li et al., 2021) | 79.0 | 8.0 | 74.8 | **5.0** | 41.0 |
| DeepAugment + AugMix (Hendrycks et al., 2020) | **75.8** | 3.9 | 53.6 | 21.2 | 18.8 |
| SIN (Geirhos et al., 2019) | **60.2** | 2.4 | **77.3** | 56.2 | **5.6** |
| **Shape-Texture Debiased Training (ours)** | 76.9 | 3.5 | 67.5 | 17.4 | 27.4 |

Table 3: Compare with state-of-the-art methods using ResNet-50 on ImageNet (IN), ImageNet-A (IN-A), ImageNet-C (IN-C), Stylized-ImageNet (S-IN), and on defending against FGSM on ImageNet. We use green to denote significant improvement, red to denote performance drop, and gray to denote similar performance. We observe our shape-texture debiased training is the **only** method that successfully leads to improvements over the vanilla baseline on all benchmarks.

| Datasets | VANILLA | S-BIASED | T-BIASED | DEBIASED |
|---|---|---|---|---|
| ImageNet-Sketch | 23.8 | 27.9 | 24.3 | **28.4** |
| ImageNet-R | 36.2 | 40.6 | 36.7 | **40.8** |
| Kylberg Texture | 99.5 | 99.1 | **99.6** | 99.5 |
| Flicker Material | 74.6 | 73.3 | **79.2** | 75.8 |

Table 4: The performance comparison between Vanilla, Shape-biased, Texture-biased, and Shape-Texture Debiased models on ImageNet-Sketch, ImageNet-R, Kylberg Texture, and Flicker Material datasets. We note the shape-biased and the shape-texture debiased models perform better on shape datasets (ImageNet-Sketch and ImageNet-R); the texture-biased and the shape-texture debiased models perform better on texture datasets (Kylberg Texture and Flicker Material).

**Comparing to SoTAs.** We further compare our shape-texture debiased model with the SoTA on ImageNet and ImageNet-A (CutMix + MoEx (Li et al., 2021)), the SoTA on ImageNet-C (DeepAugment + AugMix (Hendrycks et al., 2020)), and the SoTA on Stylized-ImageNet (SIN (Geirhos et al., 2019)). Interestingly, we note the improvements of all these SoTAs are not consistent across different benchmarks. For example, as shown in Table 3, SIN significantly improves the results on Stylized-ImageNet, but at the cost of huge performance drop on ImageNet (-16.2%) and ImageNet-C (-2.3%). Our shape-texture debiased training stands as the only method that can improve the vanilla training baseline holistically.

## 4.3 ABLATIONS

**Comparing to model ensembles.** An alternative but naïve way for obtaining the model with both shape and texture information is to ensemble a shape-biased model and a texture-biased model. We note this ensemble strategy yields a model of on-par performance with our shape-texture debiased model on ImageNet (77.2% *vs.* 76.9%). Nonetheless, interestingly, when measuring model robustness, such model ensemble strategy is inferior than ours. For example, compared to our proposed debiased training, this ensemble strategy is 1.5% worse on ImageNet-A (2.0% *vs.* 3.5%), 1.1% worse on ImageNet-C (68.6 mCE *vs.* 67.5 mCE), 1.1% worse on Stylized-ImageNet (16.3% *vs.* 17.4%), and 7.0% worse on defending against FGSM (20.4% *vs.* 27.4%). Moreover, due to model ensemble, this strategy is 2× expensive at the inference stage. These evidences clearly demonstrate the effectiveness and efficiency of the proposed shape-texture debiased training.

**Does our method help models to learn debiased shape-texture representations?** Here we take a close look at whether our method indeed prevents models from being biased toward shape or texture during learning. We evaluate models in Section 4.2 on two kinds of datasets: (1) ImageNet-Sketch dataset (Wang et al., 2019) and ImageNet-R (Hendrycks et al., 2020) for examining how well models can capture shape; and (2) Kylberg Texture dataset (Kylberg, 2011) and Flicker Material dataset (Sharan et al., 2014) for examining how well models can capture texture. Specifically, since object categories from two texture datasets are not compatible to that from ImageNet dataset, we retrain the last fc-layer (while keeping all other layers untouched) of all models on Kylberg Texture dataset or Flicker Material dataset for 5 epochs. The results are shown in Table 4.

We first analyze results on ImageNet-Sketch dataset. We observe our shape-texture debiased models are as good as the shape-biased models, and significantly outperforms the texture-biased models and the vanilla training models. For instance, using ResNet-50, our shape-texture debiased training and

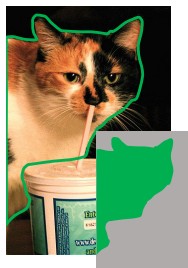 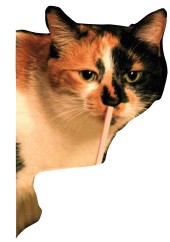 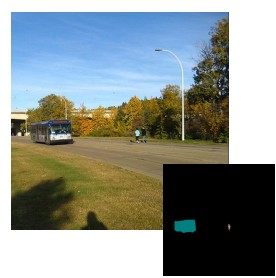 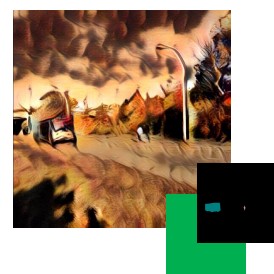

Texture Source Image & Label    Texture Source Object    Shape Source Image & Label    Generated Image & Label

Figure 5: Illustration of the data preparation pipeline of our shape-texture debiased neural network training on the semantic segmentation task.

shape-biased training achieve 28.4% top-1 accuracy and 27.9% top-1 accuracy, while texture-biased training and vanilla training only get 24.3% top-1 accuracy and 23.8% top-1 accuracy. A similar observation can be seen from ImageNet-R. These results support that our method helps models acquire stronger shape representations than the vanilla training.

We next analyze results on Kylberg Texture dataset. Similarly, we observe that our debiased model are comparable to the texture-biased model and the vanilla training model, and get better performance than the shape-biased model. On Flicker Material dataset, we observe that our debiased models are better than the vanilla training model and the shape-biased model. This phenomenon suggests texture information is effectively caught by our shape-texture debiased training. As a side note, it is expected that vanilla training are better than shape-biased training on these texture datasets, as Geirhos et al. (2019) point out that ImageNet trained models (*i.e.*, vanilla training) also tend to be biased towards texture.

With the analysis above, we conclude that, compared to vanilla training, our shape-texture debiased training successfully helps networks effectively acquire both shape and texture representations.

**Combining with other data augmentation methods.** Our shape-texture debiased neural network training can be viewed as a data augmentation method, which trains models on cue conflict images. Nonetheless, our method specifically guides the model to learn debiased shape and texture representations, which could potentially serve as a complementary feature to other data augmentation methods. To validate this argument, we train models using a combination of our method and an existing data augmentation method (*i.e.*, Mixup (Zhang et al., 2018) or CutMix (Yun et al., 2019)).

We choose ResNeXt-101 (Xie et al., 2017) as the backbone network, which reports the best top-1 ImageNet accuracy in both the Mixup paper, *i.e.*, 79.9%, and the CutMix paper, *i.e.*, 80.5%. Though building upon very strong baselines, our shape-texture debiased neural network training still leads to substantial improvements, *e.g.*, it furthers ResNeXt-101-Mixup's accuracy to 80.5% (+0.6%), and ResNeXt-101-CutMix's accuracy to 81.2% (+0.7%). Meanwhile, models' generalization also get greatly improved. For example, by combining CutMix and our method, ResNeXt-101 gets additional improvements on ImageNet-A (+1.4%), ImageNet-C (-5.9%, the lower the better) and Stylized ImageNet (+7.5%). These results support that our shape-texture debiased neural network training is compatible to existing data augmentation methods.

**Shape-texture coefficient $\gamma$.** We set $\gamma = 0.8$ in our shape-texture debiased training. This value is found via the grid search over ImageNet-200 using ResNet-18. We now ablate its sensitivity on ImageNet using ResNet-50, where $\gamma$ is linearly interpolated between 0.0 and 1.0. By increasing the value of $\gamma$, we observe that the corresponding accuracy on ImageNet first monotonically goes up, and then monotonically goes down. The sweet point can be reached by setting $\gamma = 0.7$, where ResNet-50 achieves 77.0% top-1 ImageNet accuracy. Besides, we note that by setting $\gamma \in [0.5, 0.9]$ can always lead to performance improvements over the vanilla baseline. These results demonstrate the robustness of our shape-texture debiased neural network training w.r.t. the coefficient $\gamma$.

## 4.4 SEMANTIC SEGMENTATION RESULTS

We extend our shape-texture debiased neural network training to the segmentation task. We select DeepLabv3-ResNet-101 (Chen et al., 2017) as our backbone. To better incorporate our method

with the segmentation task, the following changes are made when generating cue conflict images: (1) unlike in the classification task where the whole image is used as the texture source, we use a specific object (which can cropped from the background using the segmentation ground-truth) to provide texture information in style transfer; (2) when composing the soft label for the cue conflict image, we set the label mask from texture source as the full image (since the pattern from the texture source will fill the whole image after style transfer); and (3) we set stylization coefficient $\alpha = 0.2$ and shape-texture coefficient $\gamma = 0.95$ to prevent object boundaries from being overly blurred in style transfer. Figure 5 shows an illustration of our data preparation pipeline.

**Results.** Our shape-texture debiased training can also effectively improve segmentation models. For example, our method helps DeepLabv3-ResNet-101 achieve 77.6% mIOU, significantly beating its vanilla counterpart by 1.1%. Our method still shows advantages when compared to the $2\times$ epochs training baseline. Doubling the learning schedule of the vanilla training can only lead to an improvement of 0.2%, which is still 0.9% worse than our shape-texture debiased training. These results demonstrate the potential of our methods in helping recognition tasks in general.

## 5 RELATED WORK

**Data augmentation.** Data augmentation is essential for the success of deep learning (LeCun et al., 1998; Krizhevsky et al., 2012; Simonyan & Zisserman, 2015; Zhong et al., 2020; Cubuk et al., 2019; Lim et al., 2019; Cubuk et al., 2020). Our shape-texture debiased neural network training is related to a specific family of data augmentation, called Mixup (Zhang et al., 2018), which blends pairs of images and their labels in a convex manner, either at pixel-level (Zhang et al., 2018; Yun et al., 2019) or feature-level (Verma et al., 2019; Li et al., 2021). Our method can be interpreted as a special instantiation of Mixup which blends pairs of images at the abstraction level—images' texture information and shape information are mixed. Our method successfully guides CNNs to learn better shape and texture representations, which is an important but missing piece in existing data argumentation methods.

**Style transfer.** Style transfer, closely related to texture synthesis and transfer, means generating a stylized image by combining a shape-source image and a texture-source image (Efros & Leung, 1999; Efros & Freeman, 2001; Elad & Milanfar, 2017). The seminal work (Gatys et al., 2016) demonstrate impressive style transfer results by matching feature statistics in convolutional layers of a CNN. Later follow-ups further improve the generation quality and speed (Huang & Belongie, 2017; Chen & Schmidt, 2016; Ghiasi et al., 2017; Li et al., 2017). In this work, we follow Geirhos et al. (2019) to use AdaIN (Huang & Belongie, 2017) to generate stylized images. Nonetheless, instead of applying style transfer between an image and an artistic paintings as in Geirhos et al. (2019), we directly apply style transfer on a pair of images to generate cue conflict images. This change is vital as it enables us to provide supervisions from both shape and texture during training.

## 6 CONCLUSION

There is a long-time debate about which cue dominates the object recognition. By carefully ablate the shape-biased model and the texture-biased model, we found though biased feature representations lead to performance degradation, they are complementary to each other and are both necessary for image recognition. To this end, we propose shape-texture debiased neural network training for guiding CNNs to learn better feature representations. The key in our method is that we should not only augment training set with cue conflict images, but also provide supervisions from both shape and texture. We empirically demonstrate the advantages of our shape-texture debiased neural network training on boosting both accuracy and robustness. Our method is conceptually simple and is generalizable to different image recognition tasks. We hope our work will shed light on understanding and improving convolutional neural networks.

ACKNOWLEDGEMENT

This project is partially supported by ONR N00014-18-1-2119 and ONR N00014-20-1-2206. Cihang Xie is supported by the Facebook PhD Fellowship and a gift grant from Open Philanthropy. Yingwei Li thanks Zhiwen Wang for suggestions on figures.

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
