# OpenReview forum: "Shape-Texture Debiased Neural Network Training"
_ICLR.cc/2021/Conference — ICLR 2021 Poster_

### Official Review · AnonReviewer1 · 2020-10-26
**very simple paper, good IN results, could use more detailed analysis**

**Rating:** 6
**Confidence:** 3

**Review:**

Note that I don't follow this research area closely, this is thus more an educated impression of the paper.

The paper is extremely simple. It builds on the idea of Geirhos et al. 2019 to use style transfer to augment IN training data by mixing shape and texture information but when using a training image resulting from mixing the shape of an image with label A and the texture of an image with label B, they use 0.8*A + 0.2*B as label (which is similar to mixup/cutmix, except the merging is done using style transfer). This alone seem to provide a clear boost on IN and different variants.

My main worry/question, since Geirhos paper has had a lot of attention (as well as mixup/cutmix), is why nobody has done this yet? The paper is very clear and there don't seem to be any hidden difficulty (except maybe using an auxiliary BN?) If indeed there are no similar work, I think the clear performance boost on IN with a simple and clearly explained method justifies publication.

I would however complain the analysis is not very strong. In particular, a natural alternative to the proposed approach would be to jointly train the convolutional layers but use three different final fc layers to classify synthetic images based on shape, synthetic images based on texture, or natural images. This experiment might help to clarify why the method works.

Other comments:
- I am not sure the "debiaised" term is really appropriate
- I would prefer to see the results of 4.3 in tables (some could actually be in table 1), they would be more readable, more exhaustive and allow more effective analysis (with the current presentation I have a hard time believing the current approach peforms better than cutmix) + I would like to actually see the curve corresponding to the sensitivity to gamma
- for the segmentation results, the results are really not sufficient: what are the results with gamma=1?
- end of p.7 "which can BE cropped"
- personally, I think it's a shame one has to go to p.5 to understand what the method is actually doing when it is so simple. In general, a lot could be shortened in the first 4 pages, and dedicated to better analysis and more exhaustive evaluation

---

> ### Author Response · Authors · 2020-11-21
> **No similar work before, experiments about three fc layers and gamma=1 for segmentation**
>
> We thank the reviewer for the detailed comments and the appreciation of our work. We address the concerns below:
>
> ---
> Q1: Has anybody done any similar work before?
>
> A1: To our best knowledge, our work is indeed the first one in this direction!
>
> It is possible that not using auxiliary BN (which just published in CVPR 2020) prevents others from observing the benefits of the proposed shape-texture debiased training. For example, without auxiliary BN, the proposed method on ResNet-50 can only achieve 75.1% top-1 ImageNet accuracy, which is much lower (1.8%) than the one with auxiliary BN. We will make this part more clear for facilitating the future research in this direction.
>
> ---
> Q2: use three different final fc layers to classify training images based on shape, synthetic images based on texture, or natural images
>
> A2: Thanks for your suggestion.
>
> First, we want to clarify that, when using the CrossEntroyLoss, our method is equivalent to using three **weight-shared** fc layers to classify training images based on shape, synthetic images based on texture, or natural images. For example, as clarified in the implementation from mixup’s author (https://github.com/hongyi-zhang/mixup/issues/5), applying the cross entropy loss to the softly constructed label ($\widetilde{y}$ in Equation (1) in our paper), is equivalent to compute cross entropy loss for the shape label $y_s$ and the texture label $y_t$ separately, and then weighted average them.
>
> Given the only difference between our method and the reviewer’s suggestion is whether the fc layer should be shared or not (no other algorithmic changes), we expect the corresponding impacts will be small. Currently we are running this comparison, and will report the results once available.
>
> \###### update ######
>
> To reduce the computational cost, we first conducted the suggested experiment on ImageNet-200 using ResNet-18. As shown below, the performance gap between the weight-shared version (our proposed version) and the non-weight-shared version (the reviewer’s suggestion using 3 fully connected layers) is quite small (~0.3%). This result may suggest that using three different final fc layers in training will not have significant impacts on the final performance. We will update the results again once the experiments on ImageNet using ResNet-50 are done.
>
>
> |              Models              | Top-1 Accuracy |
> |----------------------------------|----------------|
> | ResNet-18                        |           71.3 |
> | Debiased-ResNet-18 (original)    |           73.6 |
> | Debiased-ResNet-18 (3 fc layers) |           73.3 |
>
>
> ---
> Q3: segmentation results when gamma=1?
>
> A3: The segmentation result with gamma=1 is 77.5 mIOU. We note this result is slightly worse than gamma=0.95 but much better than the vanilla training strategy (possibly due to that shape cue is crucial for the segmentation task). We will update the paper accordingly.
>
> ---
> Response to other comments:
>
> Thank you so much for the suggestions on our writing. We will polish the paper accordingly.

---

### Official Review · AnonReviewer4 · 2020-10-27
**Promising results, but could benefit from more in depth analysis.**

**Rating:** 4
**Confidence:** 4

**Review:**

Summary:

The authors propose a method to mitigate the bias towards either texture or shape, in convolutional network training. The method follows the idea from Geirhos et al (2019), but use images randomly sampled from the same dataset, instead of style transfer from paintings. Then, depending on a manually selected hyperparameter, the weights of conflicting labels are blended by weighted average of the one-hot encoding.

###################

Reasons for score

The method proposed is simple, yet effective; however, parts of the method can be more principled, for example, the sampling of textures used for style transfer and debiasing - instead of using random pairs, random sampling from certain classes or external texture or material dataset would help understand better how debiasing works.
Since the model claims that the method is debiasing the model (or can bias it towards shape or texture), it would be interesting to validate this claim on a texture database (e.g. Materials in Context, Flickr Material Database (FMD) or Describable Textures Dataset (DTD) ). The expected outcome would be - shape biased model underperforms; texture biased model, improves texture recognition. The last two datasets are small (1K images; 6K images), so it would not increase the computational overhead.

###################

Strong points of the paper:

- visualization of the cues of the two models (texture and shape), using CAM (Class Activation Mapping);
- showing the orthogonality of shape and texture-biased models (Fig. 4), with examples of tasks where each model is more accurate or less acurate.
- significant improvement on Stylized ImageNet
- usecase of the method to mitigate an adversarial attacker (FGSM).

###################

Weaker aspects:

- method is a  bit too simple (however, it is effective).
 The whole paper can be summarized as: pick a random (texture) image, style transfer and blend the labels via weighted average.
The contribution in addition to Geirhos et al seems to consist in changing the domain of the texture source, with same dataset images.

- What is the difference between the "Vanilla" training and shape biased training? My understanding from Fig. 2 and equation (1) - the two seem the same -- however they have separate columns in Table1 -- with similar results.
Please clarify the distinction.

- the method would benefit from a more principled way of choosing textures.
The proposed method consists of  (Sec 2.1., Data Generation: "... first select a pair of images from the training set uniformly at random, and then apply style transfer to blend their shape and texture".  How the algorithm decides which of the images in the pair is a texture and which one contains shape information? Can the authors make public a list of such fixed pairs, for other researchers to evaluate the methods on?
Just using style transfer (please also cite the method used for style transfer -- or if it is novel, please describe it) is very similar to the approach in Geirhos et al (2019), and it is not clear what the contribution is apart from changing the domain of the conflicting texture information.
It would be interesting to use a texture or material dataset (e.g. Materials in Context, Flickr Material Database, Describable Textures Dataset) as a source of texture for style transfer, or use class-specific textures (e.g. fur-like texture, to bias non-animal classes towards animal classes, e.g. cat, dog).

- lack of evaluation of impact on other tasks, such as transfer learning:
How does debiasing affect the features on transfer learning tasks? For example, using the features from the debiased model (pretrained) on some external dataset, e.g. with a linear classifier on top of the features.

- style transfer seems to be performed on entire image -- e.g. Figure 2 - could the authors please clarify this?
The example in Fig. 1 shows using the shape of the object, while Fig. 2  There are more recent datasets, such as MS-COCO or LVIS which provide image segmentation which could be used for training and evaluation - and also for guiding the style transfer, e.g. restricting the texture style transfer or retexturing only to the object mask.
There is an evaluation on Semantic segmentation in Sec. 4.4; however, it is not clear from Fig 5 why the background is replaced with the texture source.


- not too much information on how / why the proposed method improves against FGSM.

- Please add the benchmarks (state-of-the-art and methods compared with, e.g. Geirhos et al in Table 2). It's difficult to chase the numbers in the paper to compare the method with existing work.

---

> ### Author Response · Authors · 2020-11-21
> **“Simple but effective” is a **merit** and we add more in depth analyses (part 2)**
>
> Q5: style transfer seems to be performed on entire image, how about on objects
>
> A5: Thanks for the suggestion. For the segmentation experiments in Section 4.4, we have tried to utilize the mask to perform style transfer exclusively insides the objects. Nonetheless, we found that training quickly gets overfitted. We hypothesize that replacing texture patterns inside an object probably will introduce some artifacts (e.g., more obvious object boundaries), therefore let models exploit shortcut features in training and fail to learn features with good generalization. But we do agree that designing more sophisticated style transfer algorithms is a promising direction to further improve our method, and we leave it as a future work.
>
> ---
> Q6: the improvement against FGSM
>
> A6: We provide two possible interpretations here:
>
> 1) As argued in [1], adversarial training (which is an effective way to improve adversarial robustness)  alleviates the texture bias of standard CNNs when trained on object recognition tasks, and helps CNNs learn a more shape-biased representation. For our method, it also helps models gain more shape-representations and be less biased towards texture, which potentially explains why we see improvements for defending against the FGSM attacker.
>
> 2) As argued in previous works [2,3], adversarial examples shift the distribution of the original data. Given that our method improves models’ representation learning, it is expected that it can also improve robustness against the FGSM attacker. We will make it clear in our paper that the purpose of evaluating robustness against FGSM attack is to serve as an evidence of demonstrating models’ domain generalization ability.
>
>
>
>
> [1] Zhang, Tianyuan, and Zhanxing Zhu. "Interpreting adversarially trained convolutional neural networks." ICML, 2019
>
> [2] Jacobsen, Jörn-Henrik, et al. "Excessive invariance causes adversarial vulnerability." ICLR, 2019
>
> [3] Xie, Cihang, and Alan Yuille. "Intriguing properties of adversarial training at scale." ICLR, 2020
>
> ---
> Q7: Please add the benchmarks
> A7: Thanks for the suggestions. The comparison to Geirhos et al. 2019 and the vanilla training is shown below (will merge these results in Table 2),
>
>
> |                                  | ImageNet | ImageNet-A | ImageNet-C | S-ImageNet | FGSM |
> |----------------------------------|----------|------------|------------|------------|------|
> | Vanilla ResNet-50                |     76.4 |        2.0 |       75.0 |        7.4 | 17.1 |
> | ResNet-50 in Geirhos et al. 2019 |     76.7 |        2.3 |       73.8 |       10.4 | 21.3 |
> | Debiased-ResNet-50               |     76.9 |        3.5 |       67.5 |       17.4 | 27.4 |
>
>
>
>
>
> Besides directly comparing our method to other state-of-the-art methods, we would like to highlight that our method potentially is compatible with others, as specifically guiding CNNs to learn better shape and texture representations is largely ignored in previous works.  For example, as reported in Section 4.3, by building upon the advanced data augmentation CutMix, our method can obtain additional improvements on ImageNet-A (+1.4%), ImageNet-C (-5.9%, the lower the better) and Stylized ImageNet (+7.5%).
>
>
>
> [1] Jacobsen, Jörn-Henrik, et al. "Excessive invariance causes adversarial vulnerability." ICLR, 2019
>
> [2] Xie, Cihang, and Alan Yuille. "Intriguing properties of adversarial training at scale." ICLR, 2020
>
> [3] Taori, Rohan, et al. "Measuring robustness to natural distribution shifts in image classification." Advances in Neural Information Processing Systems 33 (2020).

---

> > ### Comment · AnonReviewer4 · 2020-11-22
> > **Thanks for the great level of details in addressing the questions.**
> >
> > First, I would like to express thanks to the authors for the level of details in addressing all the reviewers' comments and questions.
> >
> > Regarding Q3 -- thank you for the clarification with respect to the necessity of labels being from the same dataset - an interesting candidate, for future datasets could be the MetaDataset [1]
> >
> > For Q4/A4:  Kylberg Texture dataset seems a saturated dataset, i.e. all results are in the 99.+%. Flickr Material Dataset behaves as expected, i.e. shape would not be very informative, as the images are of textures/materials, covering the entire image. It is interesting to note that the debiased model performs closer to the texture than shape-biased method.
> >
> > Re. Q7: Thank you for comparing with Geirhos et al -- adding one row in Tab2. would make it easier for readers to compare results with other methods.
> > Please also include one row in the final version, with the state-of-the-art results for each of the benchmarks, i.e.
> > ImageNet  ImageNet-A 	ImageNet-C 	S-ImageNet 	FGSM
> > Vanilla ResNet-50 	76.4 	2.0 	75.0 	7.4 	17.1
> > State-of-the-art  70.5(1)   2.1(2) ....
> > ---
> > In table caption: (1) XXYYZZ et al, (2) AABBCC et al, ...
> >
> > Please also include the baseline suggested by Rev. 2 (ensemble of T- and S-), which clearly shows the advantage of the proposed method.
> >
> > [1] Meta-Dataset: A Dataset of Datasets for Learning to Learn from Few Examples, E. Triantafillou et al, ICLR 2020

---

> > > ### Author Response · Authors · 2020-11-23
> > > **Glad to see we addressed your concerns! We will update the final version based on your suggestions.**
> > >
> > > Thanks for your suggestions which help us improve the quality of this paper. We are glad to see your concerns are addressed. We additionally comment on Q3 & Q7 as below:
> > >
> > > Re Q3: Thanks for your great recommendations! This dataset seems to be a promising candidate for extending our method in the future!
> > >
> > > Re Q7: Thanks! We will definitely include the comparison to Geirhos et al and the state-of-the-art results in our final version. The table is expected to be like below (SOTA results are found via https://paperswithcode.com/):
> > >
> > > |                                    | FLOPs | #param | ImageNet | ImageNet-A | ImageNet-C | S-ImageNet | FGSM |
> > > |------------------------------------|-------|--------|----------|------------|------------|------------|------|
> > > | ResNet-50                          | 4G    | 98M    | 76.4     | 2.0        | 75.0         | 7.4        | 17.1 |
> > > | ResNet-50 in Geirhos et al. 2019   | 4G    | 98M    | 76.7     | 2.3        | 73.8       | 10.4       | 21.3 |
> > > | Ens. Shape- & Texture-Biased       | 8G    | 196M   | 77.2     | 2.0        | 68.6       | 16.3       | 20.4 |
> > > | Debiased-ResNet-50                 | 4G    | 98M    | 76.9     | 3.5        | 67.5       | 17.4       | 27.4 |
> > > | State-of-the-art (using ResNet-50) | 4G    | 98M    | 78.4 [1] | 8.4 [2]     | 60.4 [3]    | 55.8 [4]    | *    |
> > >
> > >
> > > *still literature searching
> > >
> > > [1] Yun, Sangdoo, et al. "Cutmix: Regularization strategy to train strong classifiers with localizable features." Proceedings of the IEEE International Conference on Computer Vision. 2019.
> > >
> > > [2] Li, Boyi, et al. "On Feature Normalization and Data Augmentation." arXiv preprint arXiv:2002.11102 (2020).
> > >
> > > [3] Hendrycks, Dan, et al. "The many faces of robustness: A critical analysis of out-of-distribution generalization." arXiv preprint arXiv:2006.16241 (2020).
> > >
> > > [4] Geirhos, Robert, et al. "ImageNet-trained CNNs are biased towards texture; increasing shape bias improves accuracy and robustness." ICLR, 2019

---

> ### Author Response · Authors · 2020-11-21
> **“Simple but effective” is a **merit** and we add more in depth analyses (part 1)**
>
> We first thank the reviewer for the valuable comments, which help us to improve the quality of this paper.  We address the concerns below:
>
> ---
> Q1: the method is a bit too simple & the contribution seems to consist in changing the domain of the texture source
>
> A1: FIrst of all, we would like to argue that “simple but effective” is a merit (rather than a weak point) of this paper! Given the simplicity of our method, we believe (1) it can be easily integrated into different learning frameworks (e.g., semi-supervised learning, few-shots learning) or applied to different computer vision tasks (e.g., object detection), for additional improvements; and (2) it can be easily implemented by other researchers using different deep learning libraries.
>
> Secondly, we want to clarify that, in addition to “ changing the domain of the texture source, with same dataset images.”,  our most important contribution over Geirhos et al. is the proposed label assignment strategy. As shown in our ablation study, this simple (but definitely non-trivial) strategy successfully and substantially improves model performance on several image recognition benchmarks and adversarial robustness. And conversely, if we fail to compose the labels with information from both the shape source and the texture source, the learned models will NOT yield additional improvements (or even cause performance degradation) over the vanilla training baseline.
>
> We will revise our paper accordingly to make these points more clear.
>
>
> ---
> Q2: difference between the "Vanilla" training and shape-biased training
>
> A2: The main difference between these two methods lies in the training images. For vanilla training, the training images are the original dataset (e.g., ImageNet); for shape-biased training, the training images are the original dataset and its stylized version (e.g., ImageNet + Stylized-ImageNet). We will make it clear in the paper.
>
>
> ---
> Q3: the method would benefit from a more principled way of choosing textures
>
> A3: Thanks for the suggestion. Currently, our algorithm randomly chooses images as either the shape source or the texture source in style transfer. We have trained the same model for multiple times, and find that such randomness in style transfer will not result in any training instability or have any impacts on the final model performance. The training framework is released here for the purpose of reproducing:
> https://anonymous.4open.science/r/678bd5dd-836a-45ba-b9ff-aacd007f4f89/
>
> Additionally, we would like to clarify that extending our method to use a texture dataset (e.g., Materials in Context) as a source of texture may be not applicable at the current stage. As stated in Equation (1), we need the texture dataset and the shape dataset to have a compatible label space, for constructing the new soft label. But we do agree that, with more fine-grained texture selection, it is possible to further improve the representation learning of our method. We leave it as a future direction for our work.
>
>
> ---
> Q4: lack of evaluation of impact on other tasks, such as transfer learning
>
> A4: First, we would like to clarify that our experiment on the Kylberg Texture dataset (Section 4.3) belongs to transfer learning: we only finetune the last fc-layer of the pretrained models on the Kylberg Texture dataset. The results are as shown below:
>
>
> |        | vanilla | S-biased | T-biased | Debiased |
> |--------|---------|----------|----------|----------|
> | Res50  |   99.48 |    99.09 |    99.61 |    99.53 |
> | Res101 |   99.74 |    99.61 |    99.72 |    99.79 |
> | Res152 |   99.78 |    99.63 |    99.76 |     99.80 |
>
>
> As expected, we can observe that our debiased models are comparable to the texture-biased models and the vanilla training models (as they are texture-biased), and get better performance than the shape-biased models.
>
> Additionally, we provide new results on Flickr Material Database (FMD). Similar to the experiment on the Kylberg Texture dataset, we only finetune the last fc-layer on FMD. The results are as shown below:
>
>
>
> |       | vanilla | S-biased | T-biased | Debiased |
> |-------|---------|----------|----------|----------|
> | Res50 |    74.6 |     73.3 |     79.2 |     75.8 |
>
>
>
> If the reviewer has any particular requests on the transfer learning dataset, please let us know.
>
> Besides performing additional experiments for the transfer learning tasks, we would like to highlight that our current results already show the proposed method substantially improves the models’ representation learning. For example, by training models on ImageNet (and no further fine-tuning), our method shows much better generalization on ImageNet-C, ImageNet-A and on defending against FGSM attack, than other baselines.

---

### Official Review · AnonReviewer2 · 2020-11-05
**Good paper with noticeable limitations**

**Rating:** 7
**Confidence:** 3

**Review:**

The paper tackles the problem of an existing bias in the classification networks, which makes them focus on a particular set of features (either local, "textures", or global, "shapes"). While it is clear how to train shape- and texture-biased networks (by preparing datasets with items easier to recognize using global or local features respectively), the authors claim that a combined training on both of these datasets would improve the performance of the network.

Pros.

1. The proposed method tackles an important problem and improves over some baselines (when compared to plain training or other similar regularizations, like Mixup and CutMix) on ImageNet-based datasets. Also, it can be applied to different problems like segmentation and improve the results, which shows the generality of the approach.

2. The authors try to explain the behavior of the method by providing extensive motivation and some experimental confirmations (unfortunately, they are quite limited).

3. The paper is extremely well-written and concise.

Cons.

1. The authors claim that whether or not the trained model will have bias depends on the statistics of items in the dataset: the more items easier to recognize using shape features, the more shape-biased the model will be. A simple test for that hypothesis would be to mix shape- and texture-biased datasets and train a model or their concatenation. If the authors' hypothesis is correct, this would aid in the final performance of the model, but I did not find such an experiment or any discussion of it in the experiments section. It seems to me that a major part of the performance of this method could simply come from data augmentation (basically, the same principle as described in the Mixup paper).

2. Following on the previous point, it would be really helpful to see attention visualizations for the proposed model, similar to Figure 4. Right now it is not possible to discern whether or not it has any noticeable differences from the baseline shape model in terms of attention maps, and it could reinforce the proposed explanation of the method.

3. Another important baseline that is missing is evaluating an ensemble of two models: one texture- and one shape-biased. It is a useful baseline, since it is hyperparameter-free, while the proposed method requires a grid-search to find an optimal parameter for blending the labels. Importantly, while the authors state that there is a "sweet spot" in this continuous hyperparameter, they do not propose a way to efficiently search for it other than grid-search. While they show that for the ImageNet dataset the network is quite robust to this hyperparameter, the same result may not be true for other datasets. Searching for this hyperparameter may prove to be significantly more costly than training an ensemble of two models.

In general, the proposed method itself is quite simple and has a lot of familiarities to the previous approaches, which the authors note. The main distinguishing feature of the proposed approach is the mechanism by which it works, suggested by the authors. And, in my opinion, there is not enough numerical and qualitative justification that this approach works the way it is explained.

Yet, assuming that some additional clarifications for the mechanisms behind the proposed approach will be provided in the rebuttal, I place my initial rating as "accept".

---

> ### Author Response · Authors · 2020-11-21
> **Benefits come more from **shape-texture debiased model** rather than data augmentation, and we show attention visualizations and better results than model ensemble**
>
> We thank the reviewer for the detailed comments and the appreciation of our work. We address the concerns below:
>
> ---
> Q1: the benefits may simply come from data augmentation?
>
> A1: We want to clarify that we use **exactly the same images** to train shape-biased, texture-biased or shape-texture debiased models. The only difference between them is how to assign labels (rather than how many shape/texture images are used) during training. As illustrated in Figure 2, for the shape-texture cue conflict inputs, if we assign the labels exclusively based on the shape/texture information, the learned models will be shape/texture biased; if we assign the labels by considering both shape and texture information, the learned models will enjoy both shape and texture representations. Our results show that the proposed shape-texture debiased model shows significantly better performance than other biased models (even if their training images are the same). Besides, as demonstrated in Section 4.3 & Appendix B (in the supplementary material), both our quantitative and qualitative results suggest that the proposed shape-texture debiased training strategy indeed obtains better shape and texture representations. We hope these clarifications can alleviate the reviewer’s concern on “the benefits of this method may simply come from data augmentation’.
>
>
>
> ---
> Q2: Attention visualization for the debiased model.
>
> A2: Thanks for the suggestion. We have uploaded the appendix (in the supplementary material) for including this visualization and the corresponding analysis.
>
> ---
> Q3 (a): the baseline of ensembling two models
>
> A3 (a): Thanks for the suggestion. We include the ensembling results below
>
>
> |       Model1       |        Model2        | FLOPs | #params | ImageNet | ImageNet-A | S-ImageNet | FGSM |
> |--------------------|----------------------|-------|---------|----------|------------|------------|------|
> | Debiased Res50     | -                    | 4G    | 98M     |     76.9 |        3.5 |       17.4 | 27.4 |
> | Shape-biased Res50 | Texture-biased Res50 | 8G    | 196M    |     77.2 |          2.0 |       16.3 | 20.4 |
>
>
>
>
> We note that, compared to the proposed shape-texture debiased model, ensembling a shape-biased model and a texture biased model achieve similar performance on ImageNet classification (77.2% vs. 76.9%), but much worse performance on ImageNet-A (2.0% vs. 3.5%), S-ImageNet (16.3% vs. 17.4%) and on defending against FGSM (20.4% vs. 27.4%). Moreover, this ensemble model is 2X expensive at the inference time, i.e., FLOPs have increased from 4G to 8G. These evidences clearly demonstrate the effectiveness of the proposed shape-texture debiased training. We will update this baseline in the paper accordingly.
>
>
> Q3 (b): the cost of searching shape-texture coefficient
>
> A3 (b): Firstly, we want to clarify that the budget of grid-search here is very cheap---we use the shallow ResNet-18 on the ImageNet-200 (with 100,000 images) to find the shape-texture coefficient. Besides, our experiments show that this searched hyperparameter (gamma = 0.8) can successfully transfer to a wide range of network architectures on the ImageNet dataset. These observations possibly suggest that searching this hyperparameter just could be a one-time job (for a specific visual task like classification), and can be done cheaply by using shallow models and a small subset of the dataset.
>
> We agree with the reviewer that our method can be further benefited if a more efficient search algorithm is designed. But designing such search algorithms is non-trivial and is irrelevant to the main purpose of this paper (shape-texture debiased training improves recognition models), we tend to leave it as a future work.

---

### Official Review · AnonReviewer5 · 2020-11-08
**interesting observation, simple experiments!**

**Rating:** 7
**Confidence:** 4

**Review:**

Hypothesis: Convolutional Neural Networks are biased towards either texture or shape according to the dataset used for training. If so, can we develop an approach that can do shape-texture debiased learning?

This hypothesis is in line with Geihros et al., 2019 who observed that ImageNet trained models are texture-biased.

Verification of the hypothesis: The paper qualitatively and quantitatively shows the influence of a shape-biased and a texture-biased model.

How to solve the problem?: The authors make two observations - (1). the model with shape-biased representations and the models with texture-biased representations are highly complementary to each other; and (2) being biased towards either shape cues or texture cues may hurt the performance but there exists a sweet-spot along the interpolation path of shape-texture debiased model that allows acquiring both shape and texture representations and achieve superior performance than vanilla models.

The rest of the paper consists of experiments to demonstrate this property and show analysis for different aspects of the model.

Question: One thing that I did not understand is how shape-texture debiasing can help against adversarial attacks? The authors present some quantitative analysis demonstrating it but I still don't get the rationale behind it.

---

> ### Author Response · Authors · 2020-11-21
> **Interpretations of good adversarial defense results**
>
> Thanks for your great appreciation of our work. We address the concerns below:
>
> Q: how shape-texture debiasing can help against adversarial attacks?
>
> A: We provide two possible interpretations here:
>
> 1) As argued in [1], adversarial training (which is an effective way to improve adversarial robustness)  alleviates the texture bias of standard CNNs when trained on object recognition tasks, and helps CNNs learn a more shape-biased representation. For our method, it also helps models gain more shape-representations and be less biased towards texture, which potentially benefits model robustness for defending against the FGSM attacker.
>
> 2) As argued in previous works [2,3], adversarial examples shift the distribution of the original data. Given that our method improves models’ representation learning, it is expected that it can also improve robustness against the FGSM attacker. We will make it clear in our paper that the purpose of evaluating robustness against FGSM attack is to serve as evidence of demonstrating models’ domain generalization ability.
>
>
> [1] Zhang, Tianyuan, and Zhanxing Zhu. "Interpreting adversarially trained convolutional neural networks." ICML, 2019
>
> [2] Jacobsen, Jörn-Henrik, et al. "Excessive invariance causes adversarial vulnerability." ICLR, 2019
>
> [3] Xie, Cihang, and Alan Yuille. "Intriguing properties of adversarial training at scale." ICLR, 2020

---

### Decision · Program_Chairs · 2021-01-07
**Final Decision**

**Decision:**

Accept (Poster)

**Comment:**

After the rebuttal stage, three of four reviewers recommend acceptance, and one gives a borderline score but argues they lean positive. Concerns seem well addressed; the method is simple yet effective.